# Plasmonic coffee-ring biosensing for AI-assisted point-of-care diagnostics

Kamyar Behrouzi ®[1,2] ✉, Zahra Khodabakhshi Fard[3], Chun-Ming Chen[1], Peisheng He[1,2], Megan Teng[1,2] & Liwei Lin ®[1,2] ✉

A major challenge in addressing global health issues is developing simple, affordable biosensors with high sensitivity and specificity. Significant progress has been made in at-home medical detection kits, especially during the COVID-19 pandemic. Here, we demonstrated a coffee-ring biosensor with ultrahigh sensitivity, utilizing the evaporation of two sessile droplets and the formation of coffee-rings with asymmetric nanoplasmonic patterns to detect disease-relevant proteins as low as 3 pg/ml, under 12 min. Experimentally, a protein-laden droplet dries on a nanofibrous membrane, pre-concentrating bio-markers at the coffee ring. A second plasmonic droplet with functionalized gold nanoshells is then deposited at an overlapping spot and dried, forming a visible asymmetric plasmonic pattern due to distinct aggregation mechanisms. To enhance detection sensitivity, a deep neural model integrating generative and convolutional networks was used to enable quantitative biomarker diagnosis from smartphone photos. We tested four different proteins, Procalcitonin (PCT) for sepsis, SARS-CoV-2 Nucleocapsid (N) protein for COVID-19, Carcinoembryonic antigen (CEA) and Prostate-specific antigen (PSA) for cancer diagnosis, showing a working concentration range over five orders of magnitude. Sensitivities surpass equivalent lateral flow immunoassays by over two orders of magnitude using human saliva samples. The detection principle, along with the device, and materials can be further advanced for early disease diagnostics.

Most diseases could potentially be diagnosed in much earlier stages with affordable and user-friendly biosensing platforms such as those COVID-19 test kits widely used during the global pandemic. Access to these biosensors holds the potential to elevate global health quality in terms of cost and life savings[1-10]. Previously, considerable efforts have been made to develop cost-effective biosensors for various diseases, such as SARS-CoV-2[11-16], cancer[17-22], and bloodstream infections[23-27]. The lateral flow immunoassay (LFIA) stands out as the most effective platform with good sensitivity, simplicity, and instrumentation[28-31].

On the other hand, the gold standard for biosensing at the hospital is the enzyme-linked immunosorbent assay (ELISA) as LFIA does not have a high enough sensitivity for the early diagnosis of critical diseases such as sepsis[31,32], where effective treatment at the early stage is critical. For example, sepsis (an overreaction of the immune system) is affecting about 50 million people annually to account 19% of all global deaths[33]. Current hospital practices are using ELISA for the diagnostics and it takes 2–4 h for credible results[34]. By starting the antibiotic administration just one hour earlier can increase the chance of survival by 7.6%[35] and delayed diagnostics can exacerbate the situation[36,37]. For example, the concentration of procalcitonin (PCT), a key sepsis biomarker[38], can reach around 50–500 pg/ml after 3 h from the start of infection once symptoms emerge[39]. After 12 h as most patients arrive hospitals, this number exceeds 10 ng/ml[39]. The conventional LFIA, which operates at the ng/ml level[29,39] cannot reliably

[1]Department of Mechanical Engineering, University of California, Berkeley, CA, USA. [2]Berkeley Sensor and Actuator Center (BSAC), Berkeley, CA, USA. [3]Department of Applied Science and Technology, University of California, Berkeley, CA, USA. ✉e-mail: kbehrouzi@berkeley.edu; lwlin@berkeley.edu

detect PCT at the pg/ml level for the early detection of sepsis[39]. To address the sensitivity issue, researchers have been exploring next generation LFIAs by incorporating fluorescent labeling and imaging to improve their detection limits[40–42]. For example, combining the LFIA with activated fluorescent nanoparticles, have shown a significant improvement in the sensitivity for biomarkers like SARS-CoV-2 Nucleocapsid protein (N-Protein) with a limit of detection (LOD) of 212 pg/ml[41] but it requires special instrumentation. The LOD can be further improved by the inclusion of complicated procedures and instruments, such as Raman microscopy[43]. Research works on biosensors detectable by the naked eye or a smartphone image are also progressing, such as the aggregation-induced color change plasmonic biosensor[44] and smartphone nanoparticles on-chip sensor[45]. These studies reveal a trade-off between simplicity and sensitivity. The former provides easy sample mixing and visual changes in solution color within 5 min with limited sensitivity (around 300 ng/ml for N-Protein SARS-CoV-2 detection via the naked eye). The latter achieves better sensitivity (around 250 copies/ml) for various viral infections, including Zika and HBV (Hepatitis B Virus), but requires reactions on a microfluidic device and post-processing images[45].

In general, the sensitivity of a biosensor can be enhanced by pre-concentrating biomarkers for improved signal-to-noise ratio (SNR) or, equivalently, a lower false-positive rate. For example, the polymerase chain reaction (PCR) effectively achieves the pre-concentration for DNA sensing[46,47], but the detections of proteins and small molecules do not have such a process. The indirect use of PCR to amplify the fluorescence signal resulting from DNA-attached antibodies by specific proteins, known as quantitative immune-PCR[48], represents a successful protein pre-concentration method. Another method is the single molecule array (SiMoA) by capturing proteins with magnetic particles to amplify fluorescence signals and achieve a low limits of detection (LODs) of 5 pg/ml for prostate-specific antigen (PSA) as compared to 100 pg/ml with ELISA[49]. However, these pre-concentration methods require expensive instruments and reagents.

This article introduces a biosensing scheme inspired by a natural pre-concentration mechanism induced by evaporating sessile droplets, known as the coffee-ring effect. In this approach, protein pre-concentration is combined with enhanced light-matter interaction with plasmonic gold nanoshells (GNShs) to generate naked-eye-visible signals for detecting different proteins, including the SARS-CoV-2 N-Protein, PCT as a sepsis biomarker, PSA and CEA for cancer diagnoses. The coffee-ring effect has been utilized in sensing applications through various techniques, including fluorescent imaging[50,51], Raman microscopy[52,53], electrochemical impedance spectroscopy (EIS)[54], and computationally enhanced optical imaging[55,56]. Unlike those aforementioned coffee-ring biosensors with limitations in sensitivity and/or specificity or requiring complex instrumentation[50–56], this work eliminates intricate procedures or advanced tools to achieve a very high sensitivity of 3 pg/ml (using the PSA detection as an example) through a unique combination of coffee-ring enrichment and symmetry-broken plasmonic pattern, which is further enhanced by a deep neural network toward potential applications as an at-home biosensor kit for early disease detection.

## Results

### Plasmonic coffee-ring biosensing process

The coffee-ring effect has been previously employed in protein detections using fluorescent imaging or magnetic particle trapping but signals generated by these sensors have been difficult to analyze and/or requiring complex instruments[50–56]. Here, the asymmetric plasmonic interaction through evaporation-induced flows on a thin nanofibrous membrane has been utilized to detect low-concentration proteins both qualitatively and quantitatively. It should be noted that the unique combination of the two coffee-ring interaction effect with the pre-concentration mechanism and the asymmetric plasmonic

pattern enhances the biosensing sensitivity by more than three orders of magnitude[57]. Simple yes-or-no results can be identified by the naked eye and the concentration value can be derived using the deep neural network analyses on a smartphone image. The detection process has two simple steps as shown in Fig. 1a. First, a 5 μl sample droplet is placed on the right side of a thermally treated nanofibrous membrane for the evaporation and drying process to form a coffee-ring. Second, a 2 μl plasmonic nanoparticles droplet is placed at the left side of the first droplet to interact with the dried first droplet to form an asymmetric interaction pattern after the evaporation and drying process. The resulting interaction pattern can be captured by a smartphone image and processed using a deep machine learning algorithm. The mechanism of this plasmonic coffee-ring biosensor comprises two phenomena. First, the protein concentration increases by the coffee-ring process at the edge of the droplet and forms a clear boundary. Second, the plasmonic droplet visualizes proteins by creating a dispersed 2D plasmonic pattern due to the interactions between the proteins and nanoparticles as shown in Fig. 1b. Outside the interaction region, plasmonic nanoparticles interact with each other and form large 3D aggregates due to the lack of specific proteins and an asymmetric pattern is formed in Fig. 1c. It should be noted that the coffee-ring effect enhances the second step by providing more GNShs to interact with the specific proteins already deposited at the coffee-ring of the sample droplet. This leads to an increase in the reaction rate and an improvement in the visual signal (purple color area in the figure). Figure 1d shows the experimental results after the sample droplet deposition, where analytes within the droplet accumulate mostly on the coffee-ring of the residual droplet, however, a small portion of the analytes eventually accumulates at the spreading boundary, close to the edge of the detection region. After dropping the plasmonic droplet, its residual droplet sweeps through the coffee-ring of the sample droplet and interacts with the concentrated proteins. The evaporation induced flow drives GNShs over the coffee ring boundary of the sample droplet with high protein concentration to form the asymmetric overlapping pattern in Fig. 1e (pink color in the schematic diagram and slightly dark-color pattern in the optical photo). It is also observed that the overlapping zone containing the specifically aggregated GNShs has a dispersed gradient pattern with darker color close to the coffee ring of the sample droplet as expected due to the higher sample protein concentration.

### Evaporation steps

The droplet evaporation process on the thin nanofibrous membrane has four major steps: spreading, fixed contact radius evaporation, fixed contact angle evaporation, and backward evaporation (Fig. 2). Once a droplet is placed on the membrane, the fluid spreads towards the hydrophobic barrier of the detection substrate, while a residual droplet reaches an equilibrium between the capillary force, surface tension, and gravity effects as shown in Fig. 2a. The majority of particles stay within the residual droplet and merge together at the coffee-ring by evaporation-induced flow. During the fixed contact radius evaporation step, samples are pre-concentrated within the residual droplet (see Fig. 2b). Next, during the fixed contact angle evaporation process, the residual droplet reduces its radius while keeping a fixed contact angle (less than 5°) to deposit the rest of the particles in the central part of the coffee-ring (Fig. 2c). Lastly, the remaining fluid inside the membrane evaporates from the outer ring towards the center (Fig. 2d). The typical evaporation process of a sessile droplet on a porous membrane is explained with more details in ref. 58. In general, a thin membrane is preferred as it minimizes the fluid volume within the membrane and minimizes the number of particles in the non-specific region. Moreover, nanopores inside the membrane provide more free surface for nanoparticles to attach. By optimizing the membrane properties, one could further amplify the pre-concentration effects. In Supplementary Movie (Movie S1), each of

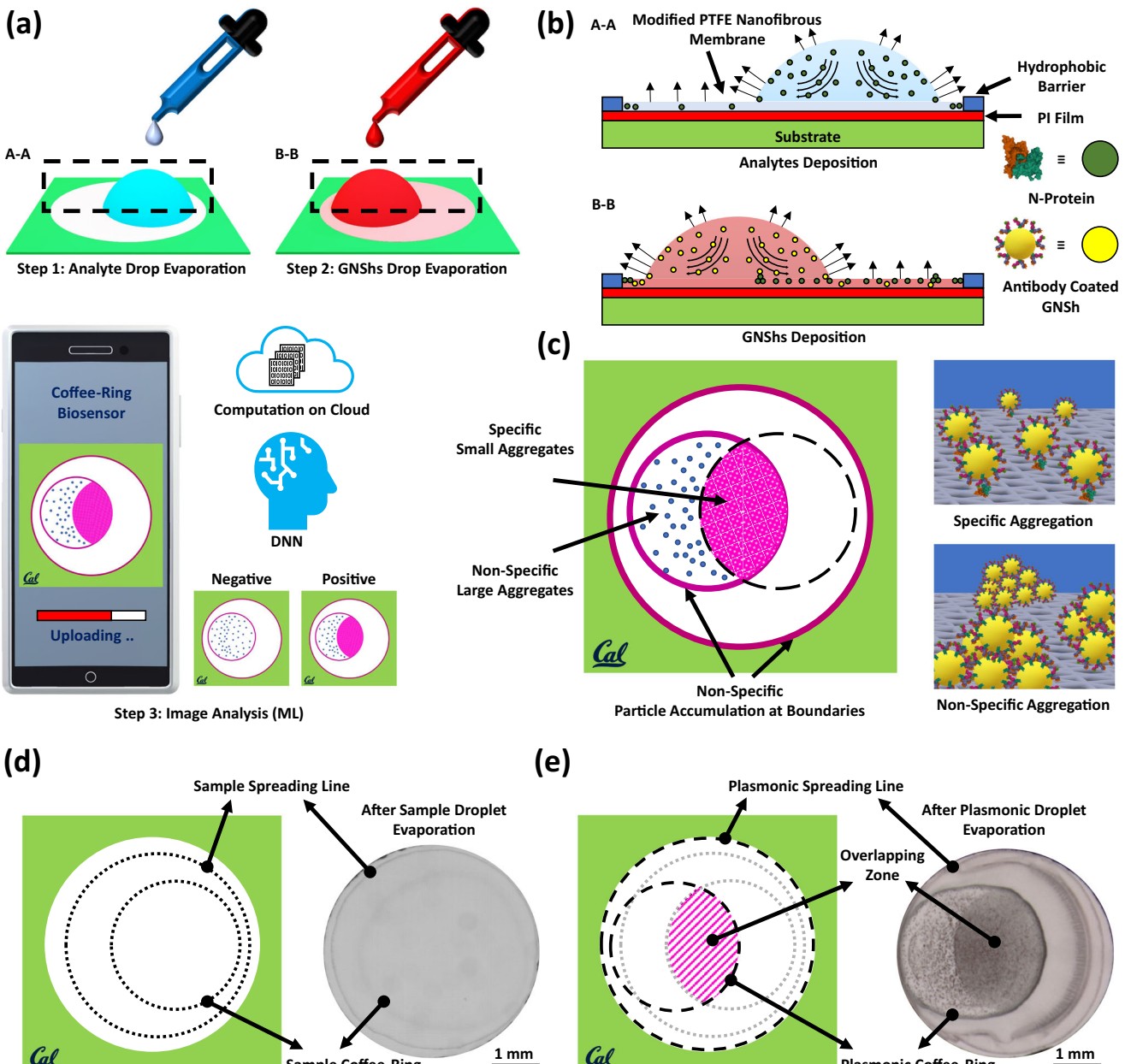

**Fig. 1 | The plasmonic coffee-ring biosensor. a** The evaporation of sample and plasmonic nanoparticles droplets on the opposite sides of the membrane, and smartphone image for biosensing. The naked eye can identify the generated positive result and a deep neural network is used to post-process the generated interaction pattern to derive the sample concentration value. The artificial intelligence-based computing can be done either on the cloud or on a local user's smartphone. **b** Evaporation induced flow within the sessile droplets pushes nanoparticles towards the boundary for pre-concentration and forms the coffee-ring. The first droplet pre-concentrates and deposits biomarkers on the membrane. The antibody-antigen interaction after the placement of the plasmonic nanoparticles droplet results in a distinguishable asymmetric pattern. **c** The plasmonic nanoparticle droplet consists of two regions: specific and non-specific zones. In the overlapping area between the two droplets, antibody conjugated GNShs are specifically interacted with proteins to generate the dispersed 2D-like pattern. Outside this specific zone, and within the plasmonic coffee-ring enclosed region, GNShs self-interactions are strong to generate large 3D aggregates and form the non-specific region. **d** Schematic diagram (left) and experimental optical photo (right) of a sample droplet after the evaporation process. The coffee-ring and initial droplet boundary are barely visible to the naked eye. **e** Schematic diagram (left) and experimental optical photo (right) of the plasmonic droplet after the evaporation process, where the sample droplet is already deposited on the substrate. The overlapping area of the two droplets shows clearly darker color due to the dispersed 2D-like pattern. The other region of the plasmonic droplet has the non-specific pattern due to the self-interactions of GNShs to generate large 3D aggregates.

the four steps are clearly shown during the plasmonic droplet evaporation process.

## Nanoparticle distribution during droplet evaporation

The particle distributions around the edge of the detection substrate and the sample coffee-ring area are visualized by using fluorescent dyes to represent proteins in the sample droplet in Fig. 2e. There are two highly visible bright lines as expected: one is at the hydrophobic boundary of the detection substrate and the other is the coffee-ring of the sample droplet due to the accumulation of the sample particles. It should be noted that the high intensity line at the coffee-ring shows the enrichment effect (see

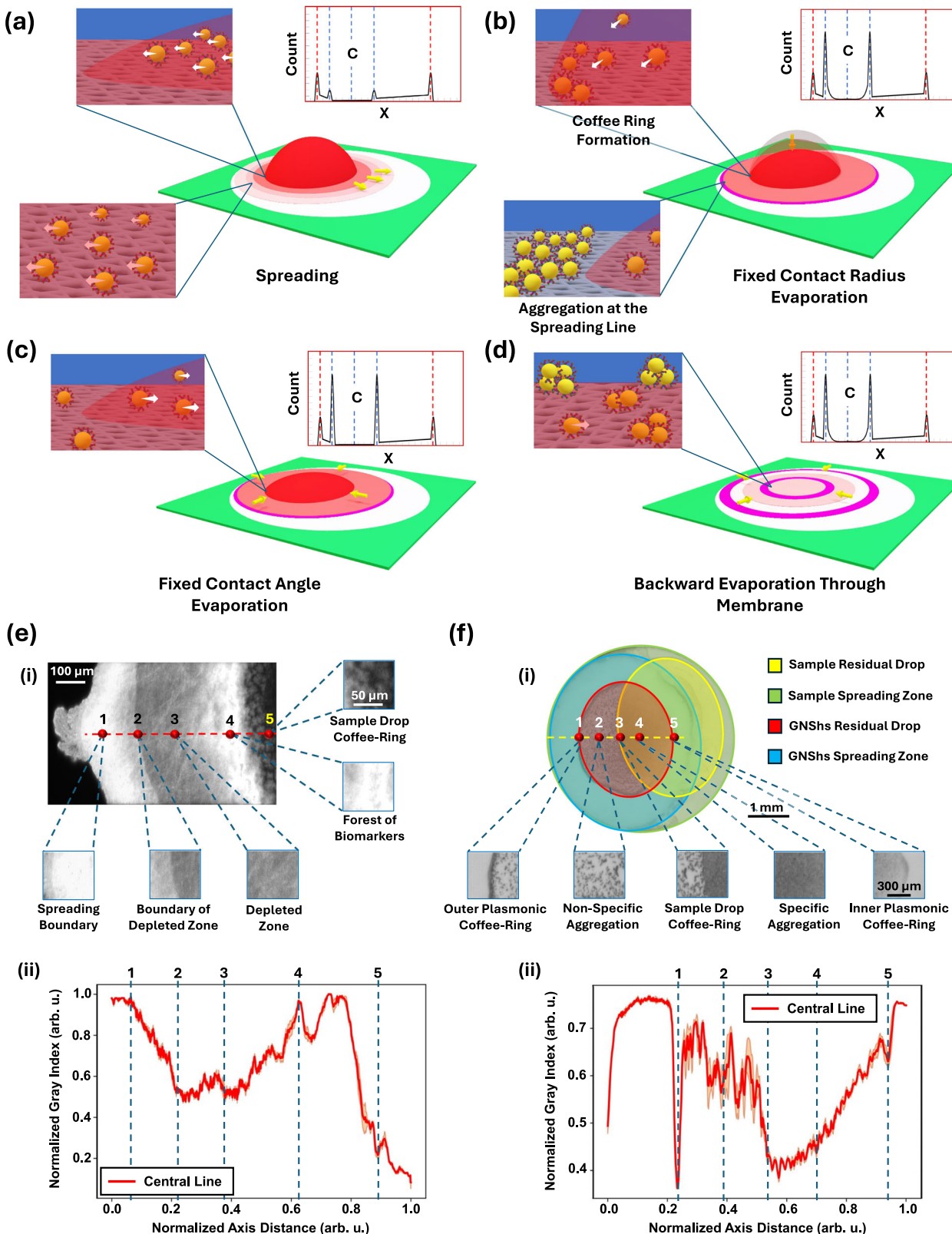

Figs. 2e, 3b, S1, S3–5). There is a particle-depleted zone with minimum particle depositions between these two boundaries. The gray index profile across the central line shows high contrast at the coffee-ring, as most particles accumulate there due to evaporation-induced flow toward the boundary of the droplet. It should be noted that biomarkers (e.g., proteins) can be trapped in the micro-crystalline residues formed by different salts inside the biological solution, such as potassium and sodium (see inset 5 of Figure 2ei). The droplet spreading process helps pushing small salt molecules towards the outer boundary, as long as the spreading process can happen (see Figure S1). If the spreading stops, accumulated salt molecules at the coffee-ring form large crystallin objects, blocking

**Fig. 2 | Evaporation steps. a** Spreading: the droplet spreads within and above the membrane and pushes particles towards the hydrophobic barrier. **b** Fixed contact radius: as evaporation begins, an internal flow develops within the residual droplet, driving particles toward the edge, forming a coffee-ring pattern. **c** Fixed contact angle: once the contract angle drops below a certain value (~ 5°), the droplet starts to shrink while maintaining this fixed angle, pulling the remaining particles towards the center. **d** Backward evaporation: The residual solution within the membrane evaporates inward, with minimal influence on the overall particle deposition pattern. Insets: particle counts where the coffee-ring positions by dashed blue lines, initial droplet boundaries by dashed red lines, and center of the residual droplet by dashed blue line plus C as center. Note, these schematic patterns are based on both our experimental data and theoretical studies[52–55]. **e** The sample droplet deposition pattern: (i) confocal microscopy image of the deposition of the sample droplet by using the fluorescent dyes. The high dye concentrations are at the spreading boundary and the coffee-ring of the residual droplet as samples are accumulated in these two places. (ii) Normalized grey index value versus axial distance of the central line passing across the image. (**f**) The GNShs droplet deposition pattern. (i) Grayscale image shows different areas on the detection zone, including residual sample droplet (yellow), sample spreading zone (green), GNShs residual drop (red), and GNShs spreading zone (blue). The asymmetric pattern at the overlapping area shows the gradual grey index value changes as the plasmonic residual droplet is pushing and interacting with the high concentration samples at the coffee ring of the residual sample droplet. (ii) Normalized grey index value for a central line passing through the plasmonic droplet coffee-ring showing the non-uniformity and the gradient in the overlapping area. Data are reported as mean values ± standard deviations (SD), based on n = 3 sample lines. It is noted that high gray index values imply less particles. These measurements are from the N-Protein at 1000 ng/ml concentration. Note that all experiments were repeated at least three times.

GNShs access to the proteins, which eventually degrades the quality of the asymmetric plasmonic pattern.

To generate the asymmetric plasmonic pattern, the plasmonic droplet is placed on the left side of the membrane. In the overlapping area between the sample droplet and the plasmonic droplet, GNShs interact with proteins to form small aggregation. In areas outside the overlapping zone, the protein concentration is low and most GNShs self-aggregate to form large non-specific aggregations as dark dots in spot # 4 of Figure 2fi. Moreover, the coffee-ring effect of the plasmonic droplet increases the local concentration of GNShs at the boundary (spots #1 and #5 in Figure 2fii), enhances the visibility of the generated plasmonic pattern. In this case, it is noted that the plasmonic droplet coffee-ring intensity at the overlapping region (spot #1 in Figure 2fii) is weaker than that of the sample droplet coffee-ring (spot # 5 in Figure 2fii), as most GNShs are attracted by proteins on the membrane in the overlapping area, such that fewer GNShs are available at the plasmonic droplet initial boundary. It is also observed that the gray index intensity level drops gradually from the coffee-ring of the plasmonic droplet to the coffee-ring of the sample droplet in the overlapping area and a transition between the specific aggregation and non-specific aggregation is clearly shown at the coffee-ring of the sample droplet in the overlapping area (spot #3 in Figure 2fii) from a continuous pattern to a noisy and granular one (Figure 2fii). The captured image profile can be used for further processing to quantify the concentration of the sample. The impact of humidity on the coffee-ring pattern has been examined. Previously, studies have shown that humidity can significantly affect droplet evaporation patterns on non-porous and non-heated substrates, particularly for smaller droplets (less than 100 μm)[59]. In our experiments, we did not observe notable variations even under extreme conditions (see Figures S22, 23). The theory of particle deposition pattern during sessile droplet evaporation has been studied extensively[58,60–62].

## Controlled coffee-ring pre-concentration through thermal treatment

The accumulation of particles during the coffee-ring process can be optimized by studying parameters such as substrate material, solution evaporation rate, droplet viscosity, and droplet volume. At the early stage of the droplet evaporation process, some nanoparticles are pushed toward the hydrophobic barrier such that some biomarkers are lost in the process. To reduce this effect, nanofibers are used in the thin hydrophilic membrane with a thermal treatment process in Fig. 3a. The substrate has five stacked materials: glass, double-sided tape, polyimide (PI) film, silicone adhesive, and a hydrophilic polytetrafluoroethylene (PTFE) nanofibrous membrane (Figure S2). Thermal flux can induce silicone nanoparticles to leave the adhesive layer onto the hydrophilic PTFE nanofibers to adjust the wettability, which keeps majority of liquid inside the residual droplet and reduces the residual droplet radius. In general, the high thermal stability of the PI film[63] makes the membrane robust during the thermal treatment process. The nanofibrous hydrophilic PTFE nanopores allow the binding of large protein clusters while permitting the passage of small molecule compounds. Furthermore, PTFE is transparent in infrared (IR) spectrum[64] for additional optical characterizations (refer to Material and Methods). For very thin diaphragms (thickness < 10 μm), key factors such as high protein absorption capacity, controlled wettability, and a high surface-to-volume ratio (e.g., using nanofibrous materials) play a crucial role in the coffee-ring biosensor. Optimizing these parameters could significantly improve the sensor's sensitivity and overall performance[65,66].

Thermal treatment is an important process to optimize the sample droplet spreading as shown in Fig. 3b. By increasing the treatment temperature below a critical value, spreading boundary moves away from the hydrophobic barrier and the majority of the sample remains inside the residual drop. However, above this critical temperature, spreading stops and salt molecules form large crystallin objects at the coffee-ring which traps the biomarkers and impedes GNShs from accessing them (Figure S3). The spreading rate (red color symbols), and maximum spreading distance (blue color symbols) versus the applied temperature between 20 °C and 160 °C are plotted in Fig. 3c. Note, spreading distance is defined as the distance between spreading boundary and the coffee-ring. It is found that as the temperature increases to 80 °C, the spreading rate and distance gradually decrease. Above 80 °C, the spreading rate and distance drop significantly and at 120 °C, nanopores are blocked and prevent the fluid movement through the membrane, leading to zero-spreading state. Figure 3d shows the comparison of the evaporation time (red color symbols) and the contact angle (blue color symbols) of the residual droplet at different temperatures. It is observed that the contact angle increases at above 80 °C for increased membrane hydrophobicity. The high temperature thermal treatment also increases the evaporation time due to the increased solution inside the residual droplet and less available surface for the evaporation.

Based on the conducted study on the influence of thermal treatment temperature on particle accumulation (Fig. 3 and Figure S4), 80 °C is chosen in this work. Although temperature sweeping experiments indicated better performance at 100 °C, the decision to work with lower temperatures was driven by the fact that we decided to conduct the sample evaporation process at 80 °C (plasmonic evaporation was conducted at room temperature), ensuring faster evaporation and minimizing the response time of our biosensor, reaching around 10 min from the beginning of the sample evaporation step. Because of the evaporation on a heated substrate, the membrane undergoes another step of thermal treatment during the sample droplet evaporation, hence, the spreading stops at lower temperatures than the critical value (120 °C), pushing the optimal temperature to the lower values. It is important to note that while higher temperatures below the boiling point of the buffer solution, phosphate-buffered

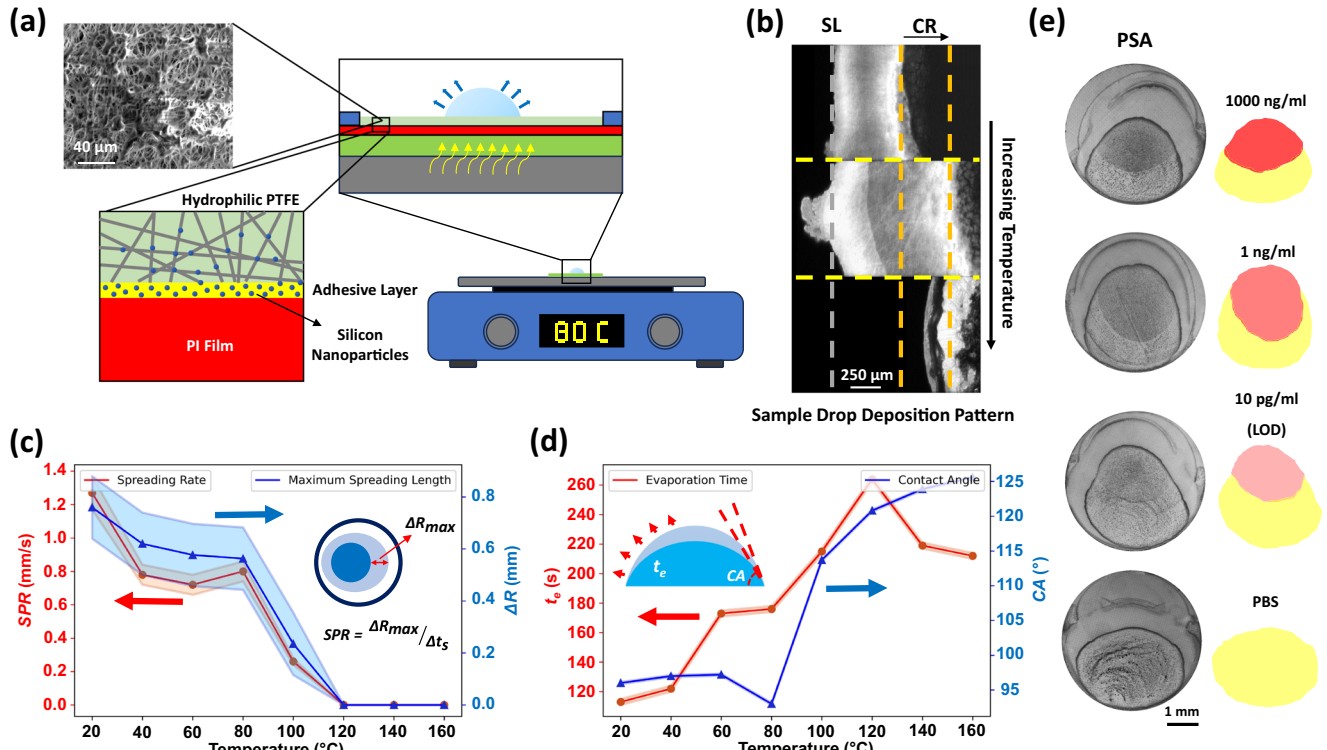

**Fig. 3 | Substrate treatment and biosensing assessment. a** The substrate is assembled in layers from bottom to top: a glass slide, double-sided tape, PI film, silicon adhesive, and a hydrophilic nanofibrous PTFE membrane (see Figure S18 for SEM). Thermal treatment modulates the membrane's wettability by evaporating silicone nanoparticles from the adhesive layer. These nanoparticles attach to the nanofibers, thereby altering the surface properties. **b** Fluorescent image of the deposited fluorescent dyes for a sample droplet deposited on the membrane under different temperatures. By increasing the temperature, the coffee-ring (CR, dashed orange line) shrinks and moves away from the spreading line (SL, dashed gray line) to enable the desirable high pre-concentration results below a critical temperature. Above the critical temperature, the formation of crystalline objects at the coffee-ring prevents GNShs to access biomarkers (SL and CR lines coalesce at the same position). **c** The distance between the spreading line and the coffee-ring is defined as the spreading variation. The spreading rate (red color symbols) and maximum spreading distance (blue color symbols) vs. applied temperature shows 80 °C is the

critical thermal treatment temperature of the prototype system as the membrane wettability decreases significantly to reduce the spreading of the droplet. **d** Experimental results of the evaporation time (red color symbols) and contact angle (blue color symbols) vs. temperature. It is found that the contact angle increases significantly above 80 °C implying more hydrophobic behavior. The evaporation time increases with the temperature, since higher thermal treatment leads to less spreading and more solution stays inside the residual droplet, leading to less available surface area for the evaporation. Results are presented as mean values ± SD, based on $n = 3$ measurements, except contact angle data which is based on the instrument repeatability. **e** The asymmetric patterns for the PSA protein. By reducing the PSA concentration, the intensity of the specific pattern decreases. Below the LOD, specific and non-specific patterns resemble the control sample (pure buffer). Key features, such as pattern intensity, gradient, higher-order statistics, and coffee-ring intensity, provide both qualitative and quantitative results. Sensing experiments were repeated at least three times.

saline (PBS), can further minimize evaporation time, such temperatures may lead to protein structure denaturation[67], affecting its affinity to the probe antibodies on the GNShs. Additionally, the heated substrate evaporation below certain temperatures[68] enhances the coffee-ring effect, as supported by our study (Figure S5).

### Coffee-ring-induced patterns and biosensing
Utilizing the optimized process, a range of proteins have been tested, including SARS-CoV-2 N-Protein, PCT as a sepsis biomarker, PSA and CEA for cancer diagnosis (Fig. 3e and Figure S6), and pure buffer with no biomarkers as a control test. Figure 3e shows the asymmetric pattern of the sample and plasmonic droplets. The color intensity of the asymmetric pattern decreases as the sample concentration drops, when fewer GNShs are attached to the proteins previously coated on the membrane. At concentrations below the limit of detection (LOD), the pattern becomes indistinguishable. The intensity of the pattern inside the overlapping area is visible to distinguish the positive versus negative results by the naked eye. An optical image captured by a smartphone can be further processed by computer vision methods to obtain additional information for the quantitative sample concentration values.

### Asymmetric plasmonic patterns
In this work, bioconjugated GNShs are attracted to biomarkers to form dispersed 2D aggregates for the smooth asymmetric plasmonic pattern (Fig. 4a). In areas with low concentration of biomarkers, GNShs interact with themselves to form large 3D aggregates for the noisy dark dots, which are visible in scanning electron microscope (SEM) images. The inherent difference between these two patterns are used to distinguish a positive detection result by the naked eye. The aggregate shape, size, and orientation under light-matter interactions all influence the nanoplasmonic effect[44] and a full numerical analysis requires substantial computational resources. Here, a simplified model based on a periodic array of GNShs with different densities is used to investigate the role of the aggregates in the optical response of plasmonic patterns. The role of geometrical properties in the optical response of nanoplasmonic aggregates have been studied previously[44,69,70]. It is worth noting that gold nanoparticles can be arranged into specific patterns to significantly enhance the system's sensitivity by several orders of magnitude; however, this approach typically requires complex instrumentation for measurements[71]. The electric field distribution of a 150 nm in diameter GNSh at its resonance frequency demonstrates strong field localization at the metal/dielectric interface

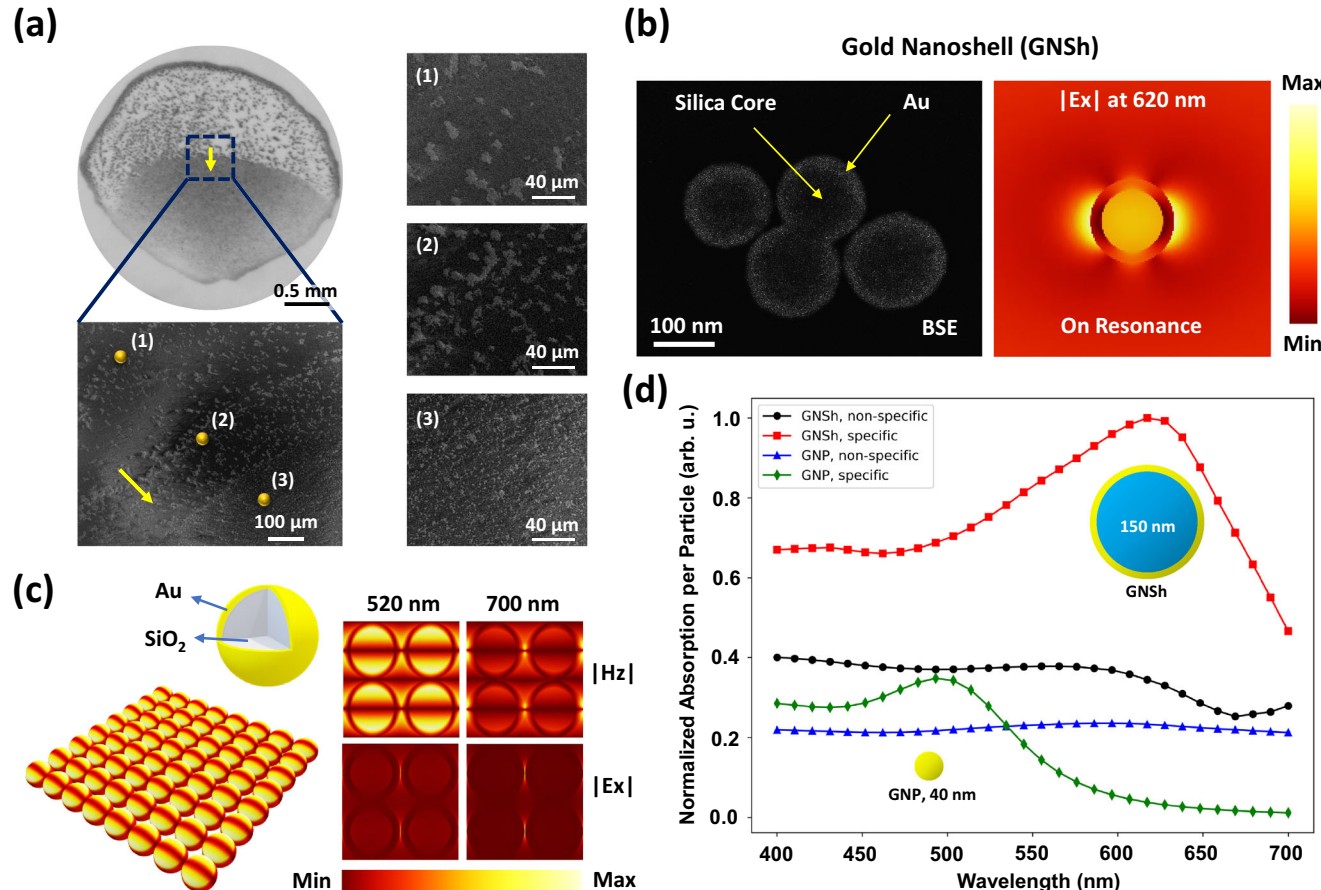

**Fig. 4 | Asymmetric plasmonic pattern. a** The specific and non-specific interactions of GNShs inside and outside of the overlapping zone. Insets: plasmonic nanoparticles inside the specific zone generate dispersed 2D-like pattern due to their interaction with coated proteins; outside the specific zone, GNShs interact with themselves because of insufficient or lack of deposited proteins and forming large aggregates. The image is the detection of N-Protein at 1000 ng/ml concentration. The yellow arrow shows the direction from the outside (point 1) to the inside (point 3) of the overlapping region. **b** SEM image of GNShs captured in the backscattered electron (BSE) mode to reveal the internal structure. Alongside is the numerically simulated electric field distribution of a 150 nm in diameter GNSh at its resonance frequency. The large particle size and high electric field localization at the metal/dielectric interface make GNShs a more effective candidate for pattern visualization when compared to those of conventional 40 nm in diameter GNPs. **c** A simplified periodic array analysis of GNShs aggregates at resonance shows that the electric field has been strongly localized in areas between the nanoparticles. **d** Optical absorption spectrum showing different spectra for specific versus non-specific aggregates. GNShs provides higher effective absorption as compared to those of GNPs. The spectra are scaled to represent per-particle absorption and subsequently normalized to the maximum value across all spectra.

(Fig. 4b) to enhance the visibility of plasmonic pattern with increased optical absorptions. The large size and high localization factor at the metal/dielectric boundary offer superior GNShs pattern visualization when compared to those of conventional 40 nm in diameter gold nanoparticles (GNPs). The GNShs are widely available, easy to synthesize, and robust plasmonic nanoparticles, making them a good choice in an at-home biosensing kit. However, many other plasmonic nanoparticles, with various shapes, sizes, and compositions, can be explored to further enhance the performance of the coffee-ring biosensor[72–74]. The simplified periodic array analysis of GNShs aggregates shows the electric field has been strongly localized in areas between the nanoparticles in Fig. 4c. Based on the simplified model, the optical absorption spectrum reveals two broadband peaks around 430 nm (in violet color range) and 620 nm (in red color range) for the dispersed GNShs aggregates, compared to 500 nm (in green color range) for the GNPs. The non-specific aggregates have a more uniform absorption across the visible spectrum as shown in Fig. 4d. Furthermore, GNShs have been compared with conventional 40 nm GNPs to show that GNShs exhibit higher optical absorption compared to those of the 40 nm GNPs, which is consistent with findings in the literature[75]. Numerical modeling has been extensively discussed in Supplementary Note (Figure S7).

## Deep neural network for protein sensing and quantification

It is found from experiments that asymmetric plasmonic patterns under relatively high biological sample concentrations are visible to the naked eye for qualitative identification without providing information for the specific concentration values. A dataset of asymmetric patterns for diverse biomarkers and control tests have been gathered in this work to train a Convolutional Neural Network (CNN) with the VGG-16 architecture (Fig. 5a)[76]. To enhance the CNN model performance and considering our limited dataset, we used transfer learning technique by adopting VGG-16 network already trained on the ImageNet dataset[76]. Details about the VGG-16 and the transfer learning process are in Supplementary Note (see Figure S8). Specifically, a Conditional Generative Adversarial Network (C-GAN) is used to extract data from the overlapping zone for sample concentration quantifications. The generative network is designed based on the U-Net network, originally created for the segmentation of biological samples[77] (Fig. 5b). To train the generator algorithm, a dataset of overlapping zone images is used as the input and manually segmented as labels. The training process is performed in tandem with the discriminator CNN network (Fig. 5c). Training these two networks together enabled the training of a C-GAN model capable of extracting the overlapping zones from images with noisy patterns not relevant to the sensing

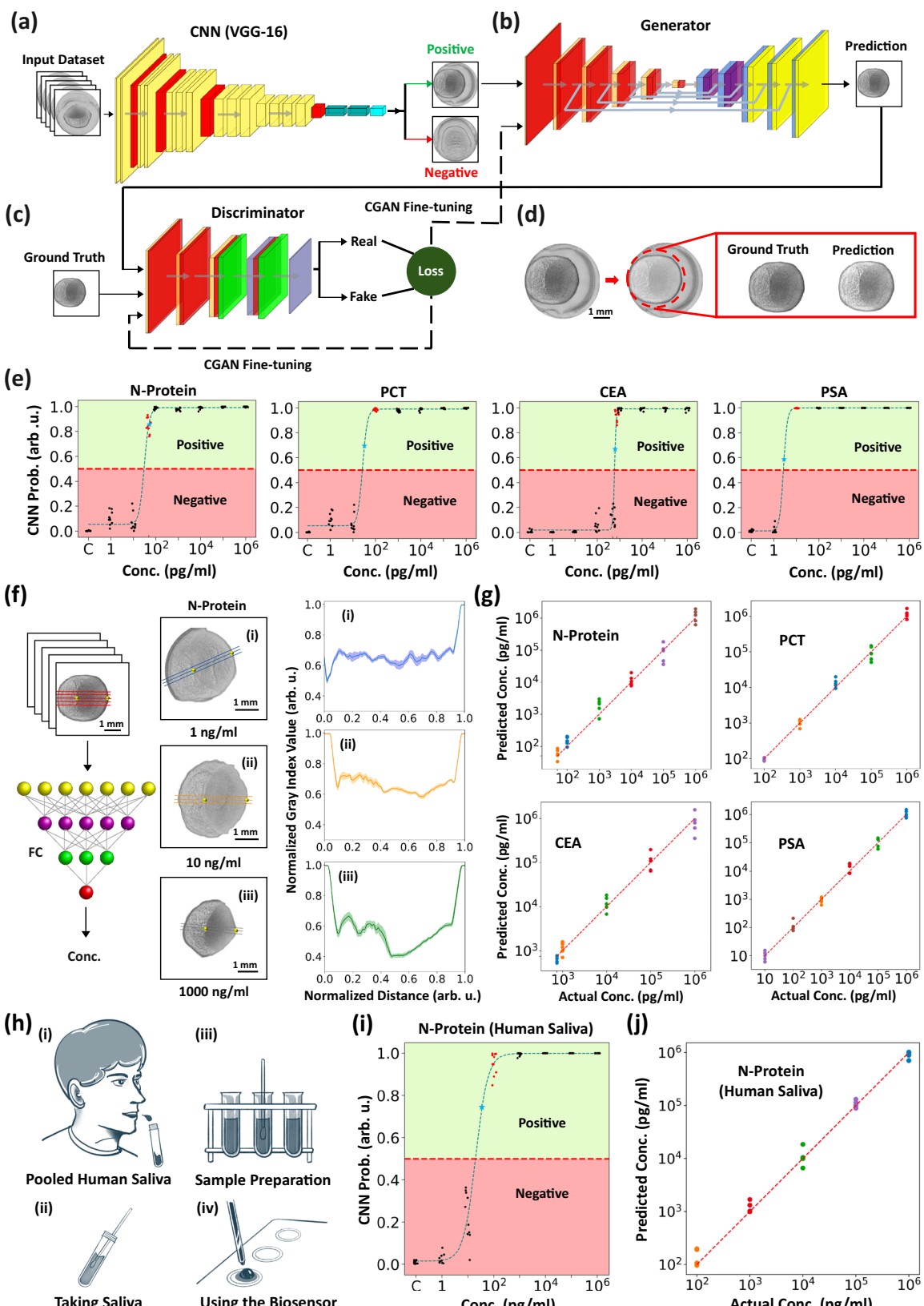

process (Fig. 5d). It should be noted that while a U-Net model alone seems to work, the literature suggests that GAN networks are robust to noise[78], which is crucial in this case since the biosensor outputs include noisy patterns that may decrease the performance of a single U-Net network. The C-GAN network in this work has been trained using the Pix2Pix method[79]. Additional information about the network and

training algorithms is included in Supplementary Note (Figures S10–S13). In Fig. 5d, the designed network has the detection zone patterns as inputs and segments out the specific zone as the output of C-GAN. Next, the extracted specific zone is fed into the image processing unit (IPU) to extract the critical points of the overlapping zone, called central points, (shown as golden spheres in Fig. 5d) for

**Fig. 5 | Deep neural network-enabled automatic biosensing. a** A CNN based on the VGG-16 architecture classifies the detection zone. It processes grayscale images of the asymmetric pattern to output the probability of a positive diagnosis (see Figure S9 for architecture). **b**, **c** A C-GAN automatically segments specific zones. The generator and discriminator are co-trained to optimize performance, with the generator producing segmentation maps resembling manually labeled images. **d** The network processes detection zone patterns and uses the C-GAN to segment the target area, removing artifacts and noise to facilitate concentration estimation. Three coffee-ring patterns from the same protein concentration are compared to demonstrate the method's consistency (Figure S27). Network structures and training details are provided in Figures S10–S13. **e** Biosensor screening performance for N-Protein, PCT, CEA, and PSA protein shows standard LODs (red points) at 50 pg/mL, 100 pg/mL, 750 pg/mL, and 10 pg/mL, respectively. Probabilistic LODs (blue stars) are ~50 pg/mL, ~30 pg/mL, ~650 pg/mL, and ~3 pg/mL (see Supplementary Note for LOD definitions). Note, the blue dashed line shows the logistic

regression fit. **f** An FC regression network predicts concentrations from crossline profiles by extracting features such as intensity, gradient, and coffee-ring patterns to establish a non-linear mapping between these features and concentration. **g** Predicted versus actual concentrations for four biomarkers (N-Protein, PCT, CEA, PSA) show excellent predictions by the FC network with minimal errors. Variations are due to testing uncertainties and protein degradation. Data are presented as mean ± SD, based on *n* = 3 crosslines. **h** The testing procedure involved spiking pooled human saliva diluted in PBS with N-Protein to create test samples. **i** Screening performance for N-Protein in saliva shows a standard LOD of 100 pg/mL and a probabilistic LOD of 50 pg/mL, outperforming equivalent LFIA tests by over two orders of magnitude (Figure S20). **j** Predicted versus actual concentrations for N-Protein mixed with human saliva closely match results obtained without saliva. All measurements were repeated at least three times. Concentration quantification was based on *n* = 5 random samples per concentration. Screening data analysis included *n* = 10 random samples.

crosslines extraction, which is being used in the next step for the concentration quantification. Note, the generator removes the artifacts and noisy features to facilitate the subsequent concentration quantification step. This network makes the sensing and quantification fully automatic.

The fine-tuned VGG-16 network has been used to analyze experimental plasmonic biosensing images of N-Protein, PCT, CEA, and PSA. Using a 0.5 probability as the boundary between positive and negative results, the screening performance of the coffee-ring biosensor for all described biomarkers are shown in Fig. 5e. Specifically, a minimum LOD of about 3 pg/ml has been achieved for PSA, which is about 30 times better than ELISA test[49]. More comparisons with other technologies are included in Table S1-2 for SARS-CoV-2 detection, and in general for smartphone-based sensing in Table S3. Notably, the observed LODs for N-Protein and PCT are well below the relevant concentrations in the corresponding diseases[39,80–82] to reveal that the sensor is suitable for early SARS-CoV-2 and sepsis diagnosis. While the observed LOD for CEA and PSA is also within relevant ranges, early cancer diagnosis requires additional biomarkers, specifically circulating tumor DNAs (ctDNAs) by highly sensitive genetic diagnostics such as the PCR method[83].

The observed variations in LODs among different proteins are due to the affinity of the antibody-antigen interactions and the quality of received proteins from vendors (refer to Supplementary Note). Apart from having low LOD values, a biosensor should respond specifically to the designed biomarker. To characterize the specificity, CEA and PSA proteins are mixed in the test as these two proteins are related to cancer diagnosis and may be present simultaneously. Experimentally, PSA-specific plasmonic droplets are applied to various droplets, including a protein-free buffer, highly concentrated (1000 ng/ml) CEA-spiked sample, PSA-spiked sample, and PSA plus a high concentration (1000 ng/ml) of CEA sample. Results show specific patterns are only found in the PSA and PSA plus CEA cases, indicating the high specificity of the coffee-ring biosensor (see Figure S9).

Each image contains information in the overlapping and non-overlapping zones within the plasmonic residual droplet, a fully connected (FC) regression network is trained based on the gray index values of the lines connecting both sides of the plasmonic residual droplet and passing through its center (Fig. 5f). This automatic algorithm can use the outputs of the C-GAN network to find the central crosslines without interfering with noisy irregular patterns outside the plasmonic coffee-ring region. By applying the trained network to four different biomarkers, the concentration quantification performance is evaluated. Results show less than half an order of magnitude deviations from expected values for most points based on a simple and affordable diagnostic method (Fig. 5g). Moreover, results show this biosensing mechanism can detect biomarkers in a large concentration variation range with 4-5 orders of magnitudes. Detailed network information is provided in Figure S14. It should be noted that, to

achieve repeatable results, we attached a 3D-printed mount to the substrate, allowing untrained users to precisely control the droplet placement locations and the gap distances between them. This setup ensures consistent asymmetric plasmonic patterns in both the calibration and test sets (see Figure S19d and S21).

Lastly, to demonstrate the capability of the coffee-ring biosensor in detecting biomarkers within complex biological samples, we tested the sensor using N-protein spiked samples mixed with pooled human saliva (see Fig. 5h–j). Our findings indicate that the coffee-ring biosensor retains its sensitivity and successfully detects N-protein within the biological solution, showing no loss in performance. Additionally, it exhibited over two orders of magnitude greater sensitivity compared to the results from equivalent LFIA tests (see Figure S20).

## Discussion
In this article, we introduced a simple, rapid (under 12 min), and affordable at-home biosensing scheme with high sensitivity and specificity, utilizing the coffee-ring effect as a natural pre-concentration process to generate asymmetric plasmonic patterns analyzed by deep learning algorithms. We combined passively controlled evaporation-induced flow within the evaporating sessile droplet with plasmonic enhanced light-matter interaction to accumulate biomarkers at the coffee-ring and visualize them. The high optical absorption of GNShs enables easy pattern visualizations. By thermally treating the nanofibrous membrane, particle flow inside the evaporating droplets has been optimized. By varying the membrane's chemical, mechanical, and geometrical properties, the coffee-ring biosensing mechanism can be further improved, while plasmonic nanoparticles and the antibody-antigen interactions can also be optimized to increase the biosensing sensitivity. The biosensing scheme is general for a wide range of biomarkers, including N-Protein of SARS-CoV-2, PCT for sepsis, as well as CEA and PSA for cancer diagnosis. An LOD as low as 3 pg/ml has been achieved for PSA, which is about three orders of magnitude less than those of conventional LFIA tests and 30 times better than ELISA tests. Furthermore, the biosensor is highly specific by testing non-specific and specific biomarkers in a single sample drop. Different deep learning networks have been developed to automate the sensing process and enhance the concentration quantification capabilities for a sensitive range of 4 to 5 orders of magnitudes. In conclusion, the biosensing mechanism presented in this work has the potential to transform the current widely accepted LFIA platforms, offering a simple and affordable diagnostic method that is sensitive enough to replace expensive and complex ELISA-based techniques, especially benefiting people in low-resource settings.

## Methods
### Materials
GNShs with an average diameter of 150 nm and a size variation of less than 15% are provided by the supplier. These nanoshells have a silica

core of 110 nm in average diameter with coated gold nanoshells of ~20 nm in thickness, extracted from high-resolution Zeiss SEM images (see, Figure S17). The elemental composition of the GNShs was examined using energy dispersive X-ray spectroscopy (EDS) analysis. All protein and polyclonal antibodies were supplied by SINO Biological as detailed below: SARS-CoV-2 N-Protein (antigen: 2019-nCoV, nucleocapsid-His recombinant protein, Cat. No. 40588-V07E, antibody: 2019-nCoV, nucleocapsid antibody, rabbit PAb, antigen affinity purified, Cat. No. 40588-T62), PCT (antigen: human CALCA / CGRP protein (His Tag), Cat. No. 13235-H08H, antibody: CALCA / CGRP antibody, rabbit PAb, antigen affinity purified, Cat. No. 13933-T16), CEA (antigen: human CEACAM-3/CD66d protein (His Tag), HPLC-verified, Cat. No. 11933-H08H, antibody: anti-CEACAM-3/CD66d antibody, rabbit polyclonal, Cat. No. 11933-T24), and PSA (antigen: human KLK3 / PSA /Kallikrein-3 protein (His Tag), Cat. No. 10771-H08H, antibody: KLK3 / Kallikrein 3 antibody, rabbit PAb, antigen affinity purified, Cat. No. 10771-RP02) proteins. Antibody purification was carried out using Millipore Amicon Ultra 0.5 ml filters. The hydrophilic PTFE nanofibrous membranes were obtained from Millipore. All reagents were purchased from Nanocomposix, unless explicitly stated otherwise.

### GNShs functionalization

We introduced 1 ml of a solution containing 5 mM potassium phosphate and 0.5% PEG with a molecular weight of 20 K to the dried NHS GNShs. This addition served to activate the NHS-Esters for subsequent antibody conjugation. Following this, the solution of GNShs was combined with purified antibodies at a specified concentration and allowed to incubate for one hour. After the incubation period, 5 μl of a 5% (w/v) hydroxylamine solution was introduced to block any unbound NHS-Esters, and the mixture was allowed to interact for an additional 10 minutes. To eliminate excess antibodies in the solution, the covalently conjugated GNShs were centrifuged three times at 3.8k RCF (relative centrifugal field). Subsequently, all non-bonded antibodies were removed, and the conjugated GNShs were supplemented with the necessary volume of a solution containing 0.1X PBS, 0.5% BSA, 0.5% Tween 20, and 0.05% Sodium Azide to achieve a final concentration of 5 μg/ml for the antibody conjugate.

### Substrate fabrication and thermal treatment process

The substrate is made by stacking a glass slide at the bottom, a double-sided tape and PI film with adhesive on top of the glass, plus a thin film of nanofibrous hydrophilic PTFE membrane on top (see Figure S2). After attaching all layers, we press the membrane with thumb finger for 10 seconds to make sure that membrane is fully in touch with the underneath silicon adhesive film. To control the wettability and surface properties of the membrane, we placed the substrate on a hot plate at a fixed temperature for 30 min. After completing the treatment process, we cooled down the temperature by applying nitrogen gas to the glass side of the substrate for about one minute.

To characterize the wettability of the thermally treated membrane, a droplet is deposited on the membrane and the system is placed on a hot plate at 80-degree Celsius (designed temperature for biosensing experiments). The droplet evaporation process is captured with a smartphone, and information is extracted to record the droplet spreading, residual droplet radius and evaporation time. Furthermore, to characterize the surface properties of the membrane, an image of buffer solution sessile droplet was captured to extract the contact angle (refer to Figure S16). The structure of the nanofibers was examined using SEM by FEI/Philips XL30.

### Sample droplet evaporation analysis

To analyze the evaporation process, we recorded the entire evaporation process using a smartphone. Recorded videos were analyzed to characterize each step in the sessile droplet evaporation process. In order to get particle deposition patterns, we mixed buffer solution with Alexa Fluor™ 488 NHS-Ester, provided by Thermo-Fisher Scientific, at 1 μg/ml concentration and used confocal microscopy to capture fluorescence images after the droplet evaporation process (Figure S4).

### Biosensing process

The biosensing process included successive evaporation of sample and GNShs droplets as depicted in Fig. 1. The sample solutions were prepared by diluting stock protein solution from 1000 ng/ml to 1 pg/ml, by a successive one-tenth dilution process. We checked the accuracy of the dilution process at high concentrations using Millipore Direct-Detect and applied the correction factor to prepared samples (concentrations multiplied by factor of 3). For each test, we heated up the hot plate to 80 °C. The biosensor performance is not very sensitive to temperature variations within 10 °C. A 5 μl sample droplet was deposited on one side of the membrane and it took about 1 min to complete the evaporation process. The system was cooled down by applying nitrogen gas to the glass side of the substrate. Next, a 2 μl of GNShs droplet was deposited on the opposite side of the sample droplet and it took about 9 minutes to complete the evaporation process. The optical image was captured using a smartphone (iPhone XR or 14) with basic settings from about 15 cm above. The sample images were taken by a tabletop microscope for better visualization and consistency.

### Biosensing in human samples

To evaluate the coffee-ring biosensor with a relevant biological sample, we used pooled human saliva from Innovative Research. A medical swab was immersed in the saliva to collect complex biomolecular mixtures to emulate the typical nasal swab collection process in the SARS-CoV-2 rapid antigen test[84]. The swab was then put into the buffer solution (PBS) and was rotated back and forth for ten times to allow the releasing and mixing of biomarkers. Next, N-protein was added to create spiked samples with specific N-protein concentrations in a mixture of non-specific proteins, enzymes, and biomolecules. The baseline measurement and sample preparation process had been previously studied using the same pooled human saliva provided by Innovative Research[85–89]. For a comparison between the performance of the coffee-ring biosensor and conventional LFIA, we applied equivalent samples to the COVID-19 antigen home test (Flowflex) at various concentrations (see Fig. 5i and S20). The background-corrected ratio of the test line to the control line (extracted using ImageJ) was used to assess the performance of the LFIA tests[90,91].

### At-home coffee-ring biosensing kit

For untrained at-home users without special tools such as a hot plate or precision pipettes, we designed a testing kit that includes glass microtubes, 3D-printed components, and a film heater (see Figure S19). A battery-powered film heater, attached to the bottom of the glass substrate, can heat up to a pre-determined temperature (i.e., 12 V generates 80 °C). To accurately handle specific volumes of the sample and plasmonic solution, precisely cut glass microtubes were used (see Figure S19c). The capillary force automatically draws the liquid in, and the length of the liquid inside the tube determines the captured volume. A 3D-printed flexible bulb, attached to a pipette tip, was assembled with the microtube to enable consistent yet simple sample handling via a droplet release mechanism. Additionally, a 3D-printed mount was placed on the top of the substrate to ensure consistent and simple droplet placement in the detection zone while maintaining consistent spacing between droplets to provide high repeatability across different tests.

It should be noted that all materials used are based on commercial products with proven stability and shelf life. The most sensitive material, the plasmonic solution, remains stable for about six months at 4 °C. It can even be dried on a separate membrane and dissolved in

the reagent solvent upon use to relax the 4 °C storage requirement, which is similar to conventional LFIAs.

## Reporting summary

Further information on research design is available in the Nature Portfolio Reporting Summary linked to this article.

## Data availability

The data generated in this study and required samples to test the code have been deposited in the Zenodo database under accession code: https://zenodo.org/records/15249376.

## Code availability

The code developed in this study has been deposited on Zenodo and can be accessed via the following link: https://zenodo.org/records/15249376.

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

## Acknowledgements

This work was supported by 2020 Seed Fund Award 2020-0000000157 from CITRIS and the Banatao Institute at the University of California, awarded to K.B. and L.L. We acknowledge Dr. Paul Lum, manager of QB3 Berkeley Nanotechnology Center (BNC) and Dr. Mary G. West, manager of QB3 Cell and Tissue Analysis Facility (CTAF) for providing instruments and materials. We acknowledge Dr. Fanping Sui for his technical support. COMSOL Multiphysics (V.6.1) license was provided by Molecular Graphics and Computation Facility (MGCF) through NIH S10OD034382.

## Author contributions

K.B. and L.L. conceived the concept; planned the project; and supervised the research. K.B. designed, fabricated, and conducted experiments and data analysis on biosensor development. K.B. and Z.K.F. conducted deep image processing and machine learning parts. K.B. and C.M.C. did the surface characterization of the thermally treated membrane. K.B. and P.H. did the evaporation analysis of sample and plasmonic droplets. K.B., M.T. and P.H. designed the at-home kit. M.T. helped with the 3D-printings. K.B., M.T. and Z.K.F. made illustrations and schemes. P.H. and K.B. captured SEM and EDS data. K.B., Z.K.F. and L.L. wrote the manuscript.

## Competing interests

The authors declare no competing interests.
