## [Transparent Peer Review file · Nature Communications]

Plasmonic Coffee-Ring Biosensing for AI-Assisted Point-of-Care Diagnostics

Corresponding Author: Dr Kamyar Behrouzi

Version 0:

Reviewer comments:

Reviewer #1

(Remarks to the Author)

In this paper, the common phenomenon of coffee ring is skillfully utilized for the enrichment of to-be-detected substances, and the detection sensitivity is high by utilizing the enhancement effect of equi-isolated excitations, and combined with machine learning algorithms, the accurate and collaborative detection of to-be-detected substances is realized; however, the data in the article is not systematic enough to validate the results in a complete way, and is not recommended to be published in this journal.

1. The article describes the results for the core-shell of gold-coated silicon and gives the thickness of the shell layer, but there are no TEM or SEM tests that can determine the thickness of the shell layer, which is an important parameter for the effect of SRES;
2. The scale of the SEM graphic in the article is too large to reflect the true shape of the interior of the coffee ring to confirm enrichment;
3. The scale of the microscope should be clearly marked, such as Fig. S6 and Fig. 5;
4. The accuracy of machine learning algorithms has a great impact on subsequent applications and it is recommended to supplement the accuracy of their predictions, for example, Fig.6 in *Science of the Total Environment* 895 (2023) 165138.
5. The performance of the substrate is poor compared to what has been reported in the literature, for example, *Nanoscale*, 2021, 13, 7574–7582, *ACS Sens.* 2023, 8, 3733–3743.

(Remarks on code availability)

Reviewer #2

(Remarks to the Author)

The authors presented a novel platform utilizing the coffee-ring effect to detect low concentrations of biomarkers relating to four different disease conditions. The novelty allows for rapid concentration and detection of these analytes with minimum-to-no sample manipulation. The overall goal is to provide an at-home test that can be interpreted with the naked eye and with the assistance of a smartphone neural network analyzer. The work can significantly impact how rapid diagnostic testing is done at the point of care. I recommend publishing after revisions are made. There are a few issues/concerns that need to be addressed.

Major comments:

1. Each plasmonic assay was tested using varying concentrations of the targeted analyte. However, only one test (PSA + CEA) was conducted using a non-targeting analyte to determine the platform's specificity. What was the specificity performance for the other assay platforms?
2. In Table S1, the authors state that the instrument used in "This Work" was "Naked Eye". However, the work also mentioned using AI to determine LoD. Can you clarify if the LoD reported is based on visualization or the AI platform? It seems to be based on the neural network analysis.
3. What were the smartphone settings used to capture the images?
4. How does the placement of the droplet within the hydrophobic barrier affect the substrate performance? Does the degree of sample/ nanoparticle ring overlap matter for the performance of the NN?
5. It is unclear why the contact angle for the GNSHs is lower than the sample on the same membrane.

Minor comments.

1. Line 160 "Insects" should be "Inset"
2. Figure 3 is missing all "i, ii, iii..." labels

(Remarks on code availability)

Reviewer #3

(Remarks to the Author)

This manuscript reports a simple and rapid method for detecting protein biomarkers by combining the coffee ring effect for sample concentration with machine learning-based image analysis of the dried coffee ring spots. One of the main advantages of this approach is that it does not rely on expensive/sophisticated instrumentation for signal/image analysis, rather, it can use optical images of the coffee ring spots, which can be captured using a smartphone camera. Experimental results show that this method is capable of detecting disease-relevant proteins, including PCT, SARS-COV-2 N protein, CEA and PSA, at clinically relevant concentrations in less than 12 minutes. While this work is of high quality, there are several major issues that diminish its significance and impact, as described below:

1. The novelty of this work is highly questionable. The approach of combining the coffee ring effect with artificial intelligence for biosensing is not a new concept and has been previously reported in several other papers:

- Hong et al., J. Biophotonics, 2020, doi: 10.1002/jbio.201960176
- Li et al., Analyst, 2020, doi: 10.1039/C9AN01624D
- Parsain et al., Food Addit. Contam. Part A Chem., 2024, doi: 10.1080/19440049.2024.2358518

Additionally, this exact concept was previously reported by the authors in the conference proceedings of Transducers 2021 (doi: 10.1109/Transducers50396.2021.9495602), which was not cited in this manuscript.

2. All of the experiments are performed in PBS, and no testing was performed using clinically relevant samples (e.g., blood, serum). Raw biofluids contain many biological and chemical components, which can interfere with the aggregation of GNShs and formation of coffee rings. Therefore, the ability of this biosensing platform to accurately detect protein biomarkers (with high sensitivity) in clinically relevant samples is questionable.

3. This platform is described as an at-home biosensor kit; however, the detection procedures require the precise handling of very small (2 μ L and 5 μ L) liquid samples, which will be challenging for ordinary people and the use of laboratory equipment (hotplate) for heating the substrate to 80°C. Furthermore, there is no data evaluating the stability and shelf life of the biosensor.

4. Images are analyzed based on the gray index values of dried coffee ring spots and therefore the accuracy of this method is highly sensitive to variations in captured images due to differences in ambient lighting, camera orientation/position with respect to the substrate and make/model of the smartphone camera. The authors should demonstrate that this approach can work in different environments (outside of the laboratory) using different types of smartphones.

In addition to these major issues, there are several concerns and questions on various technical and non-technical aspects of this work:

- The evaporation rate is strongly influenced by humidity; however, this parameter was not studied in this work. Differences in humidity can lead to variations in the coffee ring patterns. Thus, a study on the influence of humidity on the performance/accuracy of this biosensing platform should be performed.
- The sensitivity of this biosensing platform is compared to lateral flow immunoassays (LFIAs), however, this is not a fair comparison since LFIAs are much simpler to perform, less expensive, and do not require any instrumentation/hardware. It is more logical to compare this platform to other smartphone-based biosensing platforms designed for point-of-care testing, as there are many such platforms that offer comparable analytical performance.
- The use of a deep machine learning algorithm to analyze the smartphone images is performed in the cloud, which limits the utility of this platform to settings with reliable/high-speed internet connectivity.
- There are numerous machine learning algorithms and neural networks that are available for image processing and it is unclear why the authors selected the FC network and C-GAN for this work.
- What is the total size of the dataset for training the neural network? The description provided in the Materials and Methods is vague (100 unique images, which are further extended with augmentation, in this case different rotations). Typically, datasets on the order of thousands of images are required for training neural networks to achieve high accuracy. Also, I suggest including representative images (including all the applied augmentations) that were used to train the algorithm.
- It is unclear how the LOD was calculated? Typically, the LOD is calculated based on the slope of the calibration curve and the "0" concentration data, however, this data is not presented in this manuscript.
- Page 7, line 160: "Insects" is a typo.

(Remarks on code availability)

Version 1:

Reviewer comments:

Reviewer #1

(Remarks to the Author)

Although the article has been revised for some issues, it is suggested that the following issues should continue to be revised with the following comments:

1. the shell layer is not obvious and cannot be found to be a core-shell structure, which does not reflect the thickness of the gold layer, but other articles can be clearly seen, it is recommended to supplement the elemental distributions, which are used to prove the structure of the article, for example: 10.1016/j.molstruc.2024.139293 and 10.1016/j.jtice.2024.105778.
2. The method of the article has already been reported and there are already existing reports of much better performance than the literature. For example, in terms of detection concentration, the literature (doi.org/10.1016/j.bios.2021.113421) achieved 6.07 fg/mL;
3. specific examples of coffee ring boundary determination methods should be given, such as the similarity of 1 ng/mL photographs, with multiple photographs placed in the support material, thus determining that it is not a coincidence.

(Remarks on code availability)

Reviewer #2

(Remarks to the Author)

The authors made extensive revisions to the manuscript that were appropriate to strengthen the work presented and included performance analysis in human saliva to further substantiate their work. The addition of the 3D-printed droplet guide also addresses the "at-home" applicability.

Major comment:

1. In the response to the reviewer, the authors justified why the proposed coffee-ring approach is different. However, this information is missing from the introduction. The introduction only mentions LFA and lacks information about other coffee-ring papers and how the approach presented compares to those that have already been done. Such as the novelty of the presented approach that addresses gaps or improvements compared to the other coffee-ring applications. This should be added in the introduction (line 81+).

2. The sample preparation for the human saliva is a bit confusing; "Medical swabs were immersed in the saliva sample following standard procedures and then mixed with N-Protein SARS-CoV-2 spiked buffer. " It is typical for target analytes to be spiked into biofluids for a more clinical representation. However, it is unclear as to why the swab was first dipped in saliva and then dipped into the protein sample. Won't this approach lead to adsorption of the protein onto the swab? If so, was there a centrifugation step to remove the spiked saliva? If not, the current approach represents a sample that has more protein constituents than saliva and not the other way around. Details of the preparation of the saliva sample might be missing.

Also, was there a baseline saliva measurement prior to spiking to account for any intrinsic presence of the N-Protein? Details about the number of saliva tested, donor demographics, or SARS-CoV-2 history would be helpful in providing context for the baseline.

Minor comments:

1. FS10 and FS11 are a bit low resolution and hard to read. Can these figures be rotated?

(Remarks on code availability)

Answers to the Reviewers' Comments:

Reviewer 1:

1- The article describes the results for the core-shell of gold-coated silicon and gives the thickness of the shell layer, but there are no TEM or SEM tests that can determine the thickness of the shell layer, which is an important parameter for the effect of SRES:

Thanks for your careful review, we added the requested TEM images and the analysis into Figure.S17.

2- The scale of the SEM graphic in the article is too large to reflect the true shape of the interior of the coffee ring to confirm enrichment:

SEM images shown in Fig.4a is to reveal the different patterns inside and outside of the overlapping region. However, for the proof of enrichment at the coffee-ring, we showed the fluorescent images in Fig.2e,3b,S1,S3-5. We also modified Fig.4a to make the size and distribution difference clearer. To make your point clear in the manuscript, we added a short description at section “Nanoparticle Distribution During Droplet Evaporation” at page 7.

3- The scale of the microscope should be clearly marked, such as Fig. S6 and Fig. 5:

We added the scale bar to all microscopic images. The corresponding revisions applied to Fig.1, 2, 4, 5, 6, 7, S6, and S9.

4- The accuracy of machine learning algorithms has a great impact on subsequent applications and it is recommended to supplement the accuracy of their predictions, for example, Fig.6 in Science of the Total Environment 895 (2023) 165138:

We added the new components such as confusion matrices to Fig S15 in supplementary, similar to the provided paper, to address this comment. We also added the paper to the citation list.

5- The performance of the substrate is poor compared to what has been reported in the literature, for example, Nanoscale, 2021, 13, 7574–7582, ACS Sens. 2023, 8, 3733–3743:

The provided papers show a different mechanism for the plasmonic nanoparticles arrangement (making start shape), and a gold coated membrane, respectively, mostly because the authors were interested in SERS signal. However, in our case we do not incorporate any Raman measurements and only rely on visual response. Therefore, the enhancement from start shape nanoparticles would be small while making the synthesis process complex. Also the modified substrate may even hurt the proposed method, since we need special surface properties of the membrane in order to use the coffee-ring enrichment without blocking the access to the proteins of interest by covering them with crystalline objects as shown in Fig.3b, S1,S3-5.

The plasmonic nanoparticles and the substrate membrane that we chose can be optimized for improved results and these enhancements could lead to a better LOD and cleaner patterns. To

address this comment, we added the following paragraph into the manuscript about the selection criteria and cited the proposed papers as potential areas for the future investigations:

Page 8 section “Controlled Coffee-Ring Pre-Concentration through Thermal Treatment”:

“For very thin diaphragms (thickness $<10\ \mu\text{m}$), key factors such as high protein absorption capacity, controlled wettability, and a high surface-to-volume ratio (e.g., using nanofibrous materials) play a crucial role in the coffee-ring biosensor. Optimizing these parameters could significantly improve the sensor's sensitivity and overall performance [64], [65].”

Page 10 section “Asymmetric Plasmonic Patterns”:

“The GNSs are widely available, easy to synthesize, and robust plasmonic nanoparticles, making them a good choice in an at-home biosensing kit. However, many other plasmonic nanoparticles, with various shapes, sizes, and compositions, can be explored to further enhance the performance of the coffee-ring biosensor [70]–[72].”

Reviewer 2:

Thanks for reading the paper carefully and providing us with kind and informative comments. Here we addressed each of your comments:

1- Each plasmonic assay was tested using varying concentrations of the targeted analyte. However, only one test (PSA + CEA) was conducted using a non-targeting analyte to determine the platform's specificity. What was the specificity performance for the other assay platforms?

The reason that we only tested the specificity for PSA is that we wanted to have a show case, since all other aspects are general and can be applied to other analytes. The first important factor in the specificity is the antibody cross-reactivity and the second one is the non-specific attachments of existing proteins to the substrate, blocking the gold nanoshells access to the biomarkers. To demonstrate our sensor is resilient to the second factor, we repeated the N-Protein sensing mixed with human saliva samples. The experiments show the sensor is still working even under relevant biofluids. We updated Fig.5 with the new experimental data and inserted Fig.S20. The corresponding revisions in the manuscript can be found on the Materials and Methods, under section “Biosensing in Human Samples”.

2- In Table S1, the authors state that the instrument used in “This Work” was “Naked Eye”. However, the work also mentioned using AI to determine LoD. Can you clarify if the LoD reported is based on visualization or the AI platform? It seems to be based on the neural network analysis.

Thanks for your careful review, from Naked Eye we meant there is no need to use any special tool to capture the signal. However, for easy use for broad users, we developed the deep learning algorithm for the concentration quantifications. We explained this more in Table S1 caption and LOD Definitions section in Materials and Methods.

3- What were the smartphone settings used to capture the images?

To capture samples photos using smartphone we used iPhone XR and iPhone 14 with the basic settings without flashlight. We roughly took the phone in 15 cm above the samples. For better visualization, we showed images took by the tabletop electronics microscope in Fig.2 and 3. Moreover, to resolve the variations caused by the orientation of the camera we presented the rescaling algorithm in Fig.S26. We included the above-mentioned description in the Materials and Methods section.

4- How does the placement of the droplet within the hydrophobic barrier affect the substrate performance? Does the degree of sample/ nanoparticle ring overlap matter for the performance of the NN?

The first CNN model can detect the positive features from the captured images in a variety of conditions, including, the placement location and the different gap distances between droplets (see Fig.S21). However, the concentration extraction algorithm requires consistent images, meaning all deposited samples should be close to the calibration set. For this reason, we prepared a 3D-printed mount to fix the location and the gap size among all tests (see Fig.S19d). The corresponding revisions in the manuscript can be found in section “Deep Neural Network for Protein Sensing and Quantification” on page 15.

5- It is unclear why the contact angle for the GNShs is lower than the sample on the same membrane.

The reason is due to the different chemicals in GNShs droplet vs sample droplet which is mostly made of PBS. The GNShs solution was explained in the Materials and Methods section. We also added the reason behind this effect to Fig.S16.

6- Minor comments.

1. Line 160 “Insects” should be “Inset”: **Addressed**

2. Figure 3 is missing all “i, ii, iii...” labels: **Addressed, it was mislabeling, thanks for catching this. All changed to Fig.2.**

Reviewer 3:

1- The novelty of this work is highly questionable. The approach of combining the coffee ring effect with artificial intelligence for biosensing is not a new concept and has been previously reported in several other papers:

- Hong et al., J. Biophotonics, 2020, doi: 10.1002/jbio.201960176
- Li et al., Analyst, 2020, doi: 10.1039/C9AN01624D

• Parsain et al., Food Addit. Contam. Part A Chem., 2024, doi: 10.1080/19440049.2024.2358518

Additionally, this exact concept was previously reported by the authors in the conference proceedings of Transducers 2021 (doi: 10.1109/Transducers50396.2021.9495602), which was not cited in this manuscript.

We have cited the following papers on coffee-ring biosensing:

1. doi: 10.1016/j.snb.2022.132807
2. doi: 10.1039/d2ra00039c
3. doi: 10.1021/acsomega.3c03690
4. doi: 10.1021/jacs.2c06192

The focus of these papers is either on the pre-concentration mechanism or inspecting the naturally deposited patterns by using external complicated techniques such as surface-enhanced Raman spectroscopy (SERS) to detect biomarkers with limited sensitivity. The key concept in this article is to generate visually detectable specific plasmonic patterns by combining the pre-concentration effect of “two coffee-rings” with specific versus non-specific plasmonic nanoparticle interactions due to the asymmetric deposition. This unique combination and the broken symmetry allowed us to achieve sensitivities as small as 3 pg/ml, which is three orders of magnitude better than the first coffee-ring paper we published in the 2021 Transducers conference.

Our conference paper also used a different mechanism similar to lateral flow for detections. In that paper, we deposited the sample on the membrane, and the coffee-ring slightly enhanced the signal around the boundaries. A second droplet, similar to conventional LFIAs, was used to provide probe particles, such as gold nanoparticles, to visualize the test line as the target region. Comparing the LOD of the conference paper’s performance with conventional LFIAs highlights this similarity, as both have a sensitivity around 1-10 ng/ml, using equivalent samples.

The asymmetry scheme by two coffee-rings introduced in this paper along with the focus on specific versus non-specific pattern variations allowed us to achieve outstanding LODs as small as 3 pg/ml. The novelty lies in the unique combination of the coffee-ring effects with asymmetric plasmonic pattern formation. We have included the earlier conference paper from our group in the citation, along with other papers provided by the reviewer (having focus on the pre-concentration mechanism and the natural deposited patterns analysis), and added the following explanations to the manuscript:

“It should be noted that the unique combination of the two coffee-ring interaction effect with the pre-concentration mechanism and the asymmetric plasmonic pattern enhances the biosensing sensitivity by more than three orders of magnitude [56].

2- All of the experiments are performed in PBS, and no testing was performed using clinically relevant samples (e.g., blood, serum). Raw biofluids contain many biological and chemical components, which can interfere with the aggregation of GNShs and formation of coffee rings. Therefore, the ability of this biosensing platform to accurately detect protein biomarkers (with high sensitivity) in clinically relevant samples is questionable.

We are very grateful for this insightful comment. Recognizing the importance of human-derived samples, we conducted additional sensing experiments using human saliva as a case study to demonstrate the biosensor's functionality in relevant samples. Medical swabs were immersed in the saliva sample following standard procedures, then mixed with N-protein SARS-CoV-2 spiked buffer. As a benchmark, we tested the same sample with an equivalent volume using a standard OTC Covid antigen test. The results, presented in the updated Fig. 5 and newly added figure (Fig. S20), show that the Coffee-Ring biosensor sensitivity is more than two orders of magnitudes better than the equivalent LFIA tests. The corresponding revisions in the manuscript can be found on the Materials and Methods, under section “Biosensing in Human Samples”.

3- This platform is described as an at-home biosensor kit; however, the detection procedures require the precise handling of very small (2 μ L and 5 μ L) liquid samples, which will be challenging for ordinary people and the use of laboratory equipment (hotplate) for heating the substrate to 80°C. Furthermore, there is no data evaluating the stability and shelf life of the biosensor.

Thank you for mentioning this helpful comment. To create a real at-home biosensor, we replaced the hot plate with a film heater attached to the bottom of the substrate (see Fig. S19), which provides 80°C by applying 12 V. Additionally, we used glass microtubes, cut to specific lengths, to enable simple and accurate sample handling (see Fig. S19). A 3D-printed flexible bulb attached to a pipette tip was used to hold the microtube, providing a simple droplet release mechanism for untrained users. To ensure accurate and consistent droplet placement and gap control, we attached a 3D-printed mount to the assembled substrate, making the biosensing process more repeatable. All the materials that we used are based on commercial products with proven stability and shelf life. The most critical material is the plasmonic solution, which remains stable for about six months at 4°C. It can even be dried on a separate membrane and dissolved in the reagent solvent upon use, relaxing the 4°C storage condition, similar to conventional LFIAs. The corresponding revisions can be found in the manuscript under the Fig. 3 caption and in the Materials and Methods section.

4- Images are analyzed based on the gray index values of dried coffee ring spots and therefore the accuracy of this method is highly sensitive to variations in captured images due to differences in

ambient lighting, camera orientation/position with respect to the substrate and make/model of the smartphone camera. The authors should demonstrate that this approach can work in different environments (outside of the laboratory) using different types of smartphones.

The imaging process was designed in a way that users should hold the camera perpendicular to the detection zone. However, users may not have perfect control over the camera, so the captured images can be restored into their perpendicular view by scaling the hydrophobic barrier pattern into its original circular shape (see Fig S26). To investigate the effect of lighting and imaging sensor conditions, we applied different brightness (lighting condition) and noise (different imaging sensor) to the input images and tested the model again (see Fig S24-25). Our analysis shows that none of these factors can significantly affect the model, especially in the concentration estimation, as we included self-calibration, meaning the data is always being calibrated versus the background gray index value. The corresponding modification is added to Materials and Methods, under “Deep Neural Network Training” section.

5- The evaporation rate is strongly influenced by humidity; however, this parameter was not studied in this work. Differences in humidity can lead to variations in the coffee ring patterns. Thus, a study on the influence of humidity on the performance/accuracy of this biosensing platform should be performed.

Based on the literature, humidity can affect the coffee-ring evaporation process, while the effect is minimum when dealing with large droplet sizes (e.g., $200\ \mu\text{m}$ <, doi: 10.1056/NEJMoa031918). Moreover, our setup heats up the substrate and the evaporation rate is mostly dominated by the heat flux beneath the substrate, such that other factors have minimum impact. We did two series of experiments, one at different relative humidity values and the other one by using a cap over one of the samples (see Fig S22-23). Our study revealed subtle changes in the coffee-ring intensities, showing relative humidity has negligible effect on the process. Revisions reflecting this have been made to the section “Nanoparticle Distribution During Droplet Evaporation” on pages 7 and 8.

6- The sensitivity of this biosensing platform is compared to lateral flow immunoassays (LFIAs), however, this is not a fair comparison since LFIAs are much simpler to perform, less expensive, and do not require any instrumentation/hardware. It is more logical to compare this platform to other smartphone-based biosensing platforms designed for point-of-care testing, as there are many such platforms that offer comparable analytical performance.

In response to this comment, we have added Table S2 alongside Table S1. While Table S1 compares lateral flow-based biosensors with traditional gold standard techniques, Table S2 provides a comparison between our developed sensor and cutting-edge point-of-care (POC) biosensors. To ensure a fair comparison, we focused the literature review on SARS-CoV-2 diagnostics. Since the protein biomarkers vary across studies, and restricting the comparison to a single biomarker would significantly reduce the number of relevant papers, we also included multiple biomarkers in the table. Additionally, we created Table S3 to compare our coffee-ring

biosensor with other smartphone-based biosensors for various diseases, given the relatively limited number of smartphone-based biosensing technologies.

Considering the at-home coffee-ring biosensor kit, presented in Fig.S19, our proposed method is closer to LFIA than the other POC devices, therefore, we think the comparison with LFIA is reasonable. Revisions reflecting this have been made to the Deep Neural Network for Protein Sensing and Quantification section on page 14 of the manuscript.

7- The use of a deep machine learning algorithm to analyze the smartphone images is performed in the cloud, which limits the utility of this platform to settings with reliable/high-speed internet connectivity.

The reason that we mentioned cloud-based computing was to make the AI-enhanced sensing as general as possible so the user does not need to have the application. However, all the deep learning algorithms presented in this paper can be run on any smartphone without going to the cloud. We added a notice to the caption of Fig.1 to make this clear for the reader.

8- There are numerous machine learning algorithms and neural networks that are available for image processing and it is unclear why the authors selected the FC network and C-GAN for this work.

VGG and ResNet are previously shown as good candidates for similar tasks using transfer learning as shown in the reference paper: W. Wang, K. Chen, X. Ma, and J. Guo, “Artificial intelligence reinforced upconversion nanoparticle-based lateral flow assay via transfer learning,” *Fundam. Res.*, vol. 3, no. 4, pp. 544–556, 2023, doi: <https://doi.org/10.1016/j.fmre.2022.03.025>. Specifically, the VGG-16 architecture is chosen over ResNet-50 in this work for the screening step, as based on our analysis, we observed VGG-16 has better performance (see Fig S.15). Note, as our dataset was limited the transfer learning enabled us to fine-tune well-trained models such as VGG-16 (trained on 12M images) and achieve higher accuracy and robustness in differentiating between positive and negative samples.

We need to segment out plasmonic residual droplet patterns to estimate the sample concentration. One of the basic and well-known models is U-Net to perform the segmentation process, however, to make the model robust under noisy patterns outside of the residual droplet pattern we decided to choose C-GAN. The robustness of C-GAN to noise has been mentioned in the following paper: L. D. Tran, S. M. Nguyen, and M. Arai, “GAN-Based Noise Model for Denoising Real Images,” *Lect. Notes Comput. Sci. (including Subser. Lect. Notes Artif. Intell. Lect. Notes Bioinformatics)*, vol. 12625 LNCS, pp. 560–572, 2021, doi: 10.1007/978-3-030-69538-5_34. As such, we selected C-GAN in this work.

For the concentration estimation, as we needed to process the crossline patterns, we started with the basic FC network which is the basic model for 1D-signal processing. The resulted good performance with high accuracy in predicting concentrations (see Fig S15) shows that the FC model is good enough for our application. The corresponding modifications are added to Materials

and Methods under “Deep Neural Network Training” and section “Deep Neural Network for Protein Sensing and Quantification”.

9- What is the total size of the dataset for training the neural network? The description provided in the Materials and Methods is vague (100 unique images, which are further extended with augmentation, in this case different rotations). Typically, datasets on the order of thousands of images are required for training neural networks to achieve high accuracy. Also, I suggest including representative images (including all the applied augmentations) that were used to train the algorithm.

Thanks for mentioning this, we changed the CNN model architecture to VGG-16 and used transfer learning to solve the issue regarding limited dataset size. Transfer learning enables fine-tuning of well-trained models on millions of images such as ImageNet with 12M size, allowing precise classification even by having a limited dataset as mentioned in the following paper: W. Wang, K. Chen, X. Ma, and J. Guo, “Artificial intelligence reinforced upconversion nanoparticle-based lateral flow assay via transfer learning,” *Fundam. Res.*, vol. 3, no. 4, pp. 544–556, 2023, doi: <https://doi.org/10.1016/j.fmre.2022.03.025>. Based on the cited paper, we tested VGG-16 and ResNet-50 and found out VGG-16 performs better in our case. Note, as VGG-16 was already trained on ImageNet dataset, it contains many important filters within its layers, however, fine-tuning by retraining with our dataset enables us to optimize the weights of VGG-16. Moreover, we added Fig. S26 to show the augmentation and the image processing on the dataset. The corresponding modifications are added to Materials and Methods under section “Deep Neural Network Training”.

10- It is unclear how the LOD was calculated? Typically, the LOD is calculated based on the slope of the calibration curve and the “0” concentration data, however, this data is not presented in this manuscript.

Thanks for mentioning this, we have two different definitions for LOD which is as follows:

- 1- For LFIA: the relative intensities of tests and control lines were used to quantify the sample concentration. We fitted the Hill’s model to the data and defined the LOD as the location that control sample intensity + 3*standard deviation crosses the fitted curve (see Fig S20). This method is based on the following article: J. Hu *et al.*, “Rapid genetic screening with high quality factor metasurfaces,” *Nat. Commun.*, vol. 14, no. 1, p. 4486, 2023, doi: 10.1038/s41467-023-39721-w.
- 2- For Coffee-Ring biosensor: We used the CNN network (VGG-16 architecture) to detect the positive samples with all predictions that have higher than 0.5 probability as positive, considering all 10 variations that has been calculated (see Fig.5). We defined the LOD as the smallest concentration which the probability for all its instances (from 10) stays over

the 0.5 value (see Fig 5). This definition makes the LOD similar to the limit of the quantification (LOQ). This method is based on the following paper: M. S. Draz *et al.*, “Virus detection using nanoparticles and deep neural network–enabled smartphone system,” *Sci. Adv.*, vol. 6, no. 51, p. eabd5354, Dec. 2023, doi: 10.1126/sciadv.abd5354.

- 3- We also defined a new method to determine LOD for machine learning predictions based on the estimated probabilities and the observed trend. The final neural network step to classify input into the predefined classes is the sigmoid activation function, the estimated probabilities should follow the same trend among different concentrations. As such, we used the more general form of sigmoid function which is called logistic regression. Similar to the conventional definition of LOD, we defined neural network LOD as the crossing point of the fitted logistic regression curve with the constant probability of 0.5 plus three times the maximum standard deviation at the negative zone.

Corresponding revisions are added to Materials and Methods, under section “LOD Definitions” and the caption of Fig. 5.

11- Page 7, line 160: “Insects” is a typo: Thanks for catching this. Addressed.

Responses to reviewers:

Reviewer #1:

Thank you for your insightful and constructive comments. We have addressed all comments point by point.

1- the shell layer is not obvious and cannot be found to be a core-shell structure, which does not reflect the thickness of the gold layer, but other articles can be clearly seen. It is recommended to supplement the elemental distributions, which are used to prove the structure of the article, for example: 10.1016/j.molstruc.2024.139293 and 10.1016/j.jtice.2024.105778.

Revisions: We have added high resolution SEM images of the dispersed gold nanoshells (**Figure S17**) and performed post-processing to distinguish the core and gold shell structure. We verified the composition of the gold nanoshells by using energy-dispersive X-ray spectroscopy (EDS) with peaks of Au, Si, and O by using the SEM tool with the analytical mode. The relevant references have been cited and the corresponding discussion has been added to the Materials and Methods section of the manuscript. Furthermore, **Figure 4** has been updated accordingly.

2- The method of the article has already been reported and there are already existing reports of much better performance than the literature. For example, in terms of detection concentration, the literature (doi.org/10.1016/j.bios.2021.113421) achieved 6.07 fg/mL.

Revisions: We have discussed key differences of our proposed method in the first paragraph of the results and introduction section. The suggested paper (doi.org/10.1016/j.bios.2021.113421) has an excellent sensitivity by Raman microscopy with complex setup, expensive instrument, and trained technicians. Our scheme is simple yet sensitive for affordable diagnostics without expensive tool and only an at-home kit, similar to that of lateral flow tests (**Figure S19**). Furthermore, state-of-the-art techniques are compared in **Tables S1–S3**.

3- Specific examples of coffee ring boundary determination methods should be given, such as the similarity of 1 ng/mL photographs, with multiple photographs placed in the support material, thus determining that it is not a coincidence.

Revisions: We have added the more examples into the supplementary materials **Figure S27**.

Reviewer #2:

Thank you for your thoughtful and detailed feedback. We have responded to each comment individually.

Major Comments:

1- In the response to the reviewer, the authors justified why the proposed coffee-ring approach is different. However, this information is missing from the introduction. The introduction only mentions LFA and lacks information about other coffee-ring papers and how the approach presented compares to those that have already been done. Such as the novelty of the presented approach that addresses gaps or improvements compared to the other coffee-ring applications. This should be added in the introduction (line 81+).

Revisions: We have incorporated the discussion in the introduction section.

2- The sample preparation for the human saliva is a bit confusing; "Medical swabs were immersed in the saliva sample following standard procedures and then mixed with N-Protein SARS-CoV-2 spiked buffer. " It is typical for target analytes to be spiked into biofluids for a more clinical representation. However, it is unclear as to why the swab was first dipped in saliva and then dipped into the protein sample. Won't this approach lead to adsorption of the protein onto the swab? If so, was there a centrifugation step to remove the spiked saliva? If not, the current approach represents a sample that has more protein constituents than saliva and not the other way around. Details of the preparation of the saliva sample might be missing. Also, was there a baseline saliva measurement prior to spiking to account for any intrinsic presence of the N-Protein? Details about the number of saliva tested, donor demographics, or SARS-CoV-2 history would be helpful in providing context for the baseline.

Revisions: We designed the process to add the complex protein mixture from pooled human saliva purchased from a vendor with N-Protein. First, medical swabs were put into the pooled human saliva and then dipped into buffer (PBS) solutions and rotated for 10 times. This process is to emulate the sample preparation process for the LFIA COVID-19 test, where swabs are put into nose to gather specimen and later immersed in buffer fluids. Afterwards, spiked samples with specific N-Protein concentrations were made by adding N-Protein for tests. We have included a detailed description in the Materials and Methods section.

Information from the provider is used as the baseline. Furthermore, coffee-ring sensing tests for the N-Protein concentration in both PBS and pooled human saliva have shown negligible N-protein content (below the LOD of our method). This finding has been further validated by the LFIA test (see **Figure S20**). In the pooled human saliva test, the control case contained a complex mixture of biomolecules and the buffer was not spiked with N-proteins. More information about the saliva can be found in the following references:

1. E. M. McBride et al., "Rapid liquid chromatography tandem mass spectrometry method for targeted quantitation of human performance metabolites in saliva," *J. Chromatogr. A*, vol. 1601, pp. 205–213, 2019, doi: <https://doi.org/10.1016/j.chroma.2019.04.071>.
2. G. Xun, S. T. Lane, V. A. Petrov, B. E. Pepa, and H. Zhao, "A rapid, accurate, scalable, and portable testing system for COVID-19 diagnosis," *Nat. Commun.*, vol. 12, no. 1, p. 2905, 2021, doi: [10.1038/s41467-021-23185-x](https://doi.org/10.1038/s41467-021-23185-x).
3. C. D. Heaney et al., "Comparative performance of multiplex salivary and commercially available serologic assays to detect SARS-CoV-2 IgG and neutralization titers," *J. Clin. Virol.*, vol. 145, p. 104997, 2021, doi: <https://doi.org/10.1016/j.jcv.2021.104997>.

We have incorporated these into the Materials and Methods section.

Minor Comments:

1- FS10 and FS11 are a bit low resolution and hard to read. Can these figures be rotated?

Revisions: We have rotated Figures S10 and S11 to enhance the readability for the deep learning network.